# Drug Transport across Porcine Intestine Using an Ussing Chamber System: Regional Differences and the Effect of P-Glycoprotein and CYP3A4 Activity on Drug Absorption

**DOI:** 10.3390/pharmaceutics11030139

**Published:** 2019-03-21

**Authors:** Yvonne E. Arnold, Julien Thorens, Stéphane Bernard, Yogeshvar N. Kalia

**Affiliations:** 1School of Pharmaceutical Sciences, University of Geneva & University of Lausanne, CMU-1 rue Michel Servet, 1211 Geneva 4, Switzerland; Yvonne.Arnold@unige.ch; 2Debiopharm International SA, Chemin Messidor 5-7, 1006 Lausanne, Switzerland; julienthorens@hotmail.com (J.T.); stephane.bernard@debiopharm.com (S.B.); 3Present address: Pharmacie des fins, 3 rue de Frontenex, 74000 Annecy, France

**Keywords:** intestinal permeability, regional drug absorption, Ussing chamber, biorelevant media, P-gp, CYP3A4

## Abstract

Drug absorption across viable porcine intestines was investigated using an Ussing chamber system. The apparent permeability coefficients, *P_app,pig_*, were compared to the permeability coefficients determined in humans in vivo, *P_eff,human_*. Eleven drugs from the different Biopharmaceutical Classification System (BCS) categories absorbed by passive diffusion with published *P_eff,human_* values were used to test the system. The initial experiments measured *P_app,pig_* for each drug after application in a Krebs–Bicarbonate Ringer (KBR) buffer and in biorelevant media FaSSIF V2 and FeSSIF V2, mimicking fasted and fed states. Strong sigmoidal correlations were observed between *P_eff,human_* and *P_app,pig_*. Differences in the segmental *P_app,pig_* of antipyrine, cimetidine and metoprolol confirmed the discrimination between drug uptake in the duodenum, jejunum and ileum (and colon); the results were in good agreement with human data in vivo. The presence of the P-gp inhibitor verapamil significantly increased *P_app,pig_* across the ileum of the P-gp substrates cimetidine and ranitidine (*p* < 0.05). Clotrimazole, a potent CYP3A4 inhibitor, significantly increased *P_app,pig_* of the CYP3A4 substrates midazolam, verapamil and tamoxifen and significantly decreased the formation of their main metabolites. In conclusion, the results showed that this is a robust technique to predict passive drug permeability under fasted and fed states, to identify regional differences in drug permeability and to demonstrate the activity of P-gp and CYP3A4.

## 1. Introduction

Methods to predict intestinal drug absorption in humans in vivo range from purely in silico computational techniques to preclinical animal studies in vivo. In vitro tests range from relatively simple solubilization or permeation studies to more advanced systems, mimicking several steps or even the complete passage through the gastrointestinal tract [1,2,3,4,5,6,7]. High-throughput methods (e.g., PAMPA or Caco-2 cell lines) are frequently used in early, preclinical stages. The monolayer cell culture system is limited by the absence of a full physiological membrane and the specific properties of the individual cell types present in the different segments of the intestine. These models are usually used to evaluate the permeability of the API alone since they are not sufficiently robust to support either biorelevant media or “real” formulations. Indeed, to better approach the physiological conditions, mucus-producing HT29-MTX cells and M cells inducing Raji B cells were introduced into the Caco-2 monolayer [8,9,10,11,12,13]. Although in vivo studies cannot necessarily provide insight into regional differences in drug absorption, this has been attempted to be addressed in rats by the single-pass intestinal perfusion technique or, with appropriate adjustments, the closed-loop Doluisio method [14].

In order to have comprehensive permeability data for the selection of lead drug candidates and to predict the absorption in humans in vivo, it is crucial to have a physiologically and anatomically similar model that can address the following points: (i) the identification of absorption sites (crucial for modified-release formulations and poorly soluble compounds), (ii) a full physiological membrane evaluation (including mucus layer), (iii) the influence of excipients on drug permeability, (iv) drug metabolism in epithelial cells, (v) the effect of active transporters and efflux systems, and (vi) the drug/drug interactions: These will determine the relevance of the model for humans, i.e., its predictive power.

One approach is to use an Ussing chamber system, which enables transport experiments to be performed using viable intestinal tissue ex vivo [15,16,17,18,19,20,21]; tissue integrity is monitored by the continuous measurement of transepithelial resistance. Human intestine is obviously the most suitable tissue for ex vivo evaluations, and there have been some studies over the last twenty years [22,23,24,25,26,27]. Much work was done by the Drug Metabolism and Pharmacokinetics group at AstraZeneca R&D, which published a landmark paper in 2013 that compiled 15 years of data on drug absorption focusing mainly on transport across the human jejunum and colon and the effect of pre-systemic metabolism and efflux transporters [28]. However, the availability of viable human intestine is extremely limited-samples are usually obtained from patients suffering from malignancies [23,26,28,29]. To circumvent this, rat intestines have been the traditional animal model of choice for ex vivo/in vivo permeation studies [24,30,31,32]. However, it has a number of limitations, including differences in intestinal morphology and other distinct physiological differences that can make extrapolation to humans difficult [33]. Although the major drug transporters in humans and rats are in good correlation, the enzyme expression between human and rat intestines is different: For example, rat intestine displays pre-systemic cytochrome P450 activity, but this does not correlate to the activity in the human intestine [34]. In addition to the scientific issues concerning relevance, ethical concerns can also be cited with the use of rats as they have to be sacrificed for such evaluations.

Like humans, pigs are large omnivorous mammals, and porcine intestine shows greater similarities to human intestine than the intestinal tissue from other animals [33,34,35,36]. For nutritional studies, the porcine model is known to be superior to other non-primate animal models: Despite some anatomic differences, the physiology of digestion and the associated metabolic processes are much alike between humans and pigs [37]. The gross anatomical features of the GI tract of pigs and humans are similar, although the divisions between the duodenum, jejunum and ileum are not as distinct in porcine small intestine [35]. Microscopically, the intestinal villus structure and component epithelial cell types are very alike, and the pH variations of the different regions of the gastrointestinal tract in pigs and human are, again, similar [33].

At a molecular level, the metabolic activities for Phase 1 and Phase 2 enzymes in humans and pigs are closely related [34]. Porcine intestine appears to possess a high cytochrome P450 CYP3A activity, and there is more homology between human and porcine CYP3A enzymes than with those in the rat [38,39]. In both humans and pigs, P-gp is coded by a single gene (ABCB1 (ATP-binding cassette B1) or MDR1 (multidrug resistance 1)) (cf. two P-gp homologues in rats). The alignment of the porcine and human P-gp sequences resulted in a homology of 90.8%, with a high homology in the predicted transmembrane domains known to be important for substrate binding [40]. Moreover, P-gp expression was shown to increase from the proximal to distal regions in the small intestine of Yucatan micropigs—as is the case in humans [41,42]. Multidrug-resistance-associated protein (MRP2), breast cancer-resistant protein (BCRP), peptide transporter-1 (PepT1) and organic anion-transporting polypeptide (OATP) are present in porcine intestine [34].

The objective of the present study was to demonstrate that the Ussing chamber system with porcine intestine could be a useful surrogate to predict intestinal absorption in humans. Reports describing the use of porcine intestine to model drug absorption using the Ussing chamber or related systems and with diffusion cells are scarce [43,44,45,46,47,48,49,50,51,52,53,54,55], e.g., indeed, Ussing chamber investigations have only been performed on the uptake of polyphenols present in apples [43] and in coffee [44]. To our knowledge, the only detailed investigation to date into the feasibility of using porcine intestine to predict human intestinal absorption employed a diffusion cell-based system with a view to enable the development of a “medium throughput” technique [45]. 

The systematic approach employed in the present study was similar to that used by Sjöberg et al. with an extension to include the use of physiologically relevant media [28]. The specific aims were (i) to investigate the absorption of a training set of 11 molecules across viable porcine intestine from a Krebs–Bicarbonate Ringer (KBR) buffer and, with a view approaching more physiologic conditions, from the biorelevant media Fasted State Simulated Intestinal Fluid Version 2 (FaSSIF V2) and Fed State Simulated Intestinal Fluid Version 2 (FeSSIF V2) and to correlate the experimental *P_app,pig_* with *P_eff,human_* values determined in humans in vivo; (ii) to demonstrate the ability of the model to detect the regional variation in drug absorption in the different segments of the small intestine (i.e., duodenum, jejunum and ileum)—furthermore, uptake in the colon was also evaluated—(iii) to determine *P_app,pig_* of the P-gp substrates, cimetidine and ranitidine, in the presence and absence of the P-gp inhibitor, verapamil, to demonstrate that the efflux transporter retained its activity in the porcine intestine ex vivo; and likewise, (iv) to determine *P_app,pig_* of the CYP3A4 substrates, midazolam, tamoxifen and verapamil, in the presence and absence of the potent CYP3A4 inhibitor, clotrimazole, to confirm that the enzyme retained activity. Furthermore, the respective metabolites, hydroxymidazolam, N-desmethyl-tamoxifen and norverapamil, were also quantified. 

## 2. Materials and Methods 

### 2.1. Chemicals

Antipyrine, cimetidine, clotrimazole, N-desmethyl-tamoxifen hydrochloride, furosemide, hydrochlorothiazide, ketoprofen, maleic acid, (+/−)-metoprolol-(+)-tartrate, (+/−)-norverapamil hydrochloride, piroxicam, tamoxifen, terbutaline hemisulfate and (+/−)-verapamil hydrochloride 99% were purchased from Sigma-Aldrich (St. Louis, MO, USA); (+/−)-propranolol hydrochloride and ranitidine hydrochloride were obtained from Alfa Aesar GmbH & Co KG (Karlsruhe, Germany), and carbamazepine was purchased from Acros Organics (New Jersey, USA). Midazolam and α-hydroxymidazolam were purchased from Lipomed AG (Arlesheim, Switzerland), and agar, calcium chloride dihydrate, glucose hydrate, magnesium chloride hexahydrate, potassium chloride, sodium chloride, sodium hydroxide, sodium phosphate monobasic and sodium hydrogencarbonate were obtained from Hänseler AG (Herisau, Switzerland). Sodium taurocholate was purchased from Prodotti Chimici e Alimentari S.p.A., (Basaluzzo, Italy) and lecithin (grade EPCS > 98% phospholipids) was obtained from Lipoid GmbH (Ludwigshafen, Germany).

### 2.2. Porcine Intestinal Tissue

Porcine intestinal tissue from 6-month-old female Swiss noble pigs (weight: 100–120 kg) was supplied by two local abattoirs (Abattoir de Meinier; Meinier, Switzerland and Abattoir de Loëx; Bernex, Switzerland) and was collected immediately after slaughter. In order to remove the luminal debris, the tissue was rinsed with ice-cold KBR (120 mM NaCl, 5.5 mM KCl, 2.5 mM CaCl_2_, 1.2 mM MgCl_2_, 1.2 mM NaH_2_PO_4_, 20 mM NaHCO_3_ and 11 mM glucose; pH 7.4) [56]. During transport from the slaughterhouse to the laboratory, the tissue was stored in ice-cold KBR and constantly bubbled with a 95% O_2_/5% CO_2_ gas mixture (PanGas AG; Dagmersellen, Switzerland). 

Once in the laboratory, the intestinal tissue was prepared using previously published protocols [46,57]. Briefly, the intestine was opened along the mesenteric border and rinsed with ice-cold KBR. The muscle layer was carefully removed using a scalpel and fine forceps. The remaining tunica mucosa and submucosa were cut into segments of approximately 1.5 cm^2^. Areas including Peyer’s patches were avoided. During the whole procedure, which took 5–10 min, the tissue was stored in ice-cold KBR and constantly bubbled with a gas mixture of 95% O_2_/5% CO_2_. 

### 2.3. Ussing Chamber Setup and Procedures for Intestinal Absorption Experiments 

A six Ussing chamber system coupled to a VCC MC6 MultiChannel Voltage–Current Clamp (Physiologic Instruments; San Diego, CA, USA) with a heating block and six input modules with integral dummy membranes was used for the permeation studies. A circulating water bath (ED-5, Julabo GmbH, Seelbach, Germany) was used to regulate the temperature. The Ussing chambers were set up using the method reported by Neirinckx et al. [46]. First, the Ag/AgCl electrodes were put in tips containing a congealed mixture of 3% agar in 3 M KCl. Then, the electrodes were inserted into the Ussing chambers, and the donor and acceptor compartments were filled with preheated KBR (38°C). The buffer solution was constantly bubbled with a 95% O_2_/5% CO_2_ gas mixture; in addition to oxygenating the tissue, this ensured mixing and circulation of the buffer in the two compartments. Any voltage difference between the electrodes and the transepithelial electrical resistance due to the buffer solution was eliminated. The chambers were emptied, the intestinal tissue was mounted on the sliders with an exposed surface area of 1.26 cm^2^ and the sliders were inserted into the Ussing chambers. The intestine was mounted in the Ussing chamber 45 min after harvesting from the animal. KBR (7 mL) was added to the donor and acceptor compartments and left to equilibrate for 30 min, at which point both the donor and acceptor compartments were emptied and the acceptor phase was replaced with the same volume of fresh KBR (7 mL) so as to minimize the potential impact of endogenous material released during the 30 min equilibration period.

The composition of the solution in the donor compartment was dependent on the experiments: (i) in the first study into passive drug absorption, each API from the training set of 11 molecules (Table 1) was dissolved in KBR (7 mL) to prepare a 100 µM solution; (ii) in the second series of experiments, which investigated the effect of using fasted state biorelevant media on drug absorption, each API (100 µM) was dissolved in FaSSIF V2 (7 mL; 68.62 mM NaCl, 34.8 mM NaOH, 19.12 mM maleic acid, 3 mM Na taurocholate and 3 mM lecithin; pH 6.5) [58]; (iii) in the third series, which investigated the effect of fed state conditions on drug uptake, the API was dissolved in FeSSIF V2 (7 mL; 125.5 mM NaCl, 69.9 mM NaOH, 55.02 mM maleic acid, 10 mM Na taurocholate, 2 mM lecithin, 0.8 mM glycerol monooleate and 0.8 mM sodium oleate; pH 5.8) [58].

For the experiments investigating the effect of P-gp, verapamil hydrochloride (a known P-gp inhibitor) was added to the formulation in the donor compartment, again at 100 µM and similar to the concentration reported in the literature [60]. For the study investigating the activity of CYP3A4 present in the intestinal membrane, the experiments were performed in the presence/absence of clotrimazole—a potent CYP3A4 inhibitor (*K_i_* = 18 nM)—again at 100 µM [61]. Given the experimental setup, all experiments were performed in sextuplicate.

The cumulative drug permeation across the intestinal epithelium was determined by taking aliquots (400 µL) from the acceptor compartment every 20 min (t = 20, 40, 60, 80, 100 and 120 min); the volume removed was replaced with a fresh buffer. During the experiment, the viability of the intestinal tissue was monitored by measuring the variation of the voltage during the intermittent application of a 50 µA current pulse (duration 200 ms) applied every minute. Using Ohm’s law, the transepithelial electrical resistance was calculated and used as a measure for tissue viability. Preliminary studies were performed to define the threshold transepithelial electrical resistance value below which the tissue integrity was considered to be impaired or not viable. Based on these measurements, tissues with a transepithelial electrical resistance below 15 Ω.cm^2^ were considered not to be viable or intact and were not used for the calculation of the permeability coefficients. 

Upon completion of the experiment (t = 120 min), in addition to the sample from the acceptor, a 400 µL aliquot was withdrawn from the donor compartment. The intestinal slices were cut into small pieces and extracted for 6 h using the mobile phase used for the UHPLC-MS/MS analytical method (see below). This enabled the amount of API retained in the intestinal tissue to be determined. Prior to analysis, all samples were centrifuged for 10 min at 14 000 rpm using an Eppendorf Centrifuge 5804 (Vaudaux-Eppendorf AG; Schönenbuch, Switzerland).

### 2.4. Analytical Methods

The samples were analyzed using UHPLC-MS/MS. The system consisted of a Waters ACQUITY UPLC^®^ core system and a Waters XEVO^®^ TQ-MS tandem quadrupole mass spectrometer (Milford, MA, USA). Chromatographic separation was achieved using an ACQUITY UPLC^®^ BEH C18 column, 1.7 µm, 25 × 2.1 mm, attached to an ACQUITY UPLC^®^ BEH C18 Van Guard™ Pre-column, 1.7 µm, 5 × 2.1 mm. Tandem mass spectrometry was performed in the multiple reaction monitoring (MRM), mode and the majority of the APIs (Appendix A) were analyzed with positive ion electrospray ionization (ESI). Furosemide, hydrocholorothiazide, ketoprofen and piroxicam were analyzed with negative ESI (Appendix A). The complete details of the isocratic UHPLC-MS/MS methods are reported in the Appendix A. Data acquisition was done using the MassLynx^™^ software, version 4.1.

### 2.5. Data Analysis

#### 2.5.1. Permeability Calculations

The apparent permeability coefficient for transport across the porcine intestinal tissue (*P_app,pig_*) was calculated using the following equation:(1)$$P_{app,\;pig}=\frac{dc}{dt}\times \frac{V}{A\times C_{0}}\;\left(\frac{cm}{s}\right)$$
where *dc/dt* is the change in the acceptor concentration calculated from the slope of the concentration–time curve between 20 and 80 min, *V* is the buffer volume in the donor compartment, *A* is the exposed surface area (1.26 cm^2^) and *C*_0_ is the initial concentration of the API in the donor compartment [62].

#### 2.5.2. Data Fitting

The experimental permeability values *P_app,pig_* were fitted to the in vivo permeability *P_eff,human_* values using a four parameter logistic equation (SigmaPlot software, version 12.5—Systat Software Inc.; San Jose, CA, USA) as used by Sjöberg et al., where *y*_0_ is the minimum value of *P_eff,human_*, *X*_50_ is the *P_app,pig_* when *P_eff,human_* is at half the maximum value, *a* is a scaling factor and *b* is the slope factor [28].
(2)$$P_{eff,\;human}=y_{0}+\frac{a}{1+{\left(\frac{P_{app,\;pig}}{X_{50}}\right)}^{b}}$$

#### 2.5.3. Evaluation of the Relative Contributions of Drug Deposition and Permeation During Intestinal Transport: The Transport Index (TI) 

Conventional approaches to evaluate intestinal drug absorption use the permeability coefficient as the reference parameter. This does not take into account drug retention in the membrane. The quantification of drug deposition in addition to permeation is routinely carried out in investigations into the transport of drugs across other biological membranes [63,64,65,66]. In this context, Miyake et al. recently introduced the concept of the transport index (*TI*) to reflect the sum of the amounts accumulated in the intestine (*Q_DEP_*) and permeated across the tissue (*Q_PERM_*) as a percentage of the amount applied in the donor compartment [27,67]. This is analogous to a “delivery efficiency”, which is again used in topical and transdermal delivery studies to indicate the fraction of drugs delivered from a formulation into or across the skin [68,69,70].

In the conditions used in the present study, *Q_DEP_* and *Q_PERM_* are calculated as follows:(3)$$Q_{DEP}=\frac{m_{int\;2h}}{m_{donor\;0h}}\times 100\;\left(\%\right)$$
(4)$$Q_{PERM}=\frac{m_{acc\;2h}}{m_{donor\;0h}}\times 100\;\left(\%\right)$$
where *m*_*int* 2*h*_ and *m*_*acc* 2*h*_ represent the amounts deposited in and permeated across the intestine at 2 h and *m*_*donor* 0*h*_ is the amount present in the donor compartment at *t* = 0.

### 2.6. Statistical Analysis

The data were expressed as the mean ± SD. The results were evaluated statistically using analysis of variance (one-way ANOVA) followed by Bonferroni’s multiple comparisons test or Student’s *t*-test. The level of significance was fixed at α = 0.05.

## 3. Results and Discussion

### 3.1. Intestinal Absorption of Drugs from KBR and FaSSIF V2 and FeSSIF V2

The first part of the study involved the validation of the setup. This was done by determining the *P_app,pig_* of 11 drugs from the four BCS categories (seven high and four low permeability) formulated in KBR and in FaSSIF V2 and FeSSIF V2 followed by a comparison to the *P_eff,human_* reported in the literature (Table 1 and Table 2). The mean initial transepithelial resistance of the jejunum, 41.77 ± 13.78 Ω.cm^2^ (*n* = 155), was similar to the resistance of the human intestine (duodenum/jejunum; 34 ± 12 Ω.cm^2^) [28], and its monitoring for the duration of the experiment reported on the tissue viability in the presence of KBR and, importantly, the effect of biorelevant media in the donor compartment. The ability to use biorelevant media that simulate more physiological conditions is a major advantage of this system since it enables a better approximation of the food effects on drug permeation [71]; this can be difficult with Caco-2 cells due to the cytotoxic effects [72] (necessitating the development of more complex systems [13,73,74,75]). As seen with other ex vivo models, the *P_app,pig_* values were, in general, smaller than the effective permeability *P_eff,human_* determined in humans in vivo with the Loc-I-Gut single-pass perfusion technique [76]. This can be explained by the differences between the physiological and experimental conditions. For example, *P_app,pig_* (and the apparent permeability coefficients with other species) calculated from Ussing chamber data ex vivo are derived from the drug concentration gradient observed in the acceptor compartment as the amount permeated across a small piece of intestine with a defined area gradually increasing with time. They do not take into account the amount of drug that is retained in the intestinal tissue, which can be significant for certain molecules (Table 1). In contrast, *P_eff,human_* calculated in vivo using the Loc-I-Gut technique is dependent upon the difference in concentration over a given length of tissue: (5)$$P_{eff,\;human}=\frac{Q_{in}}{A}\;ln\left(\frac{C_{out}}{C_{in}}\right)$$
where *Q_in_* is the input rate, *C_in_* and *C_out_* are the drug concentrations at the start and end of the intestinal segment and *A* is the surface area for absorption. Thus, any process contributing to the loss of the drug from the intestinal fluid will contribute to an increase in *P_eff,human_*. The surface area for absorption in the Loc-I-Gut model has typically been modelled as a smooth, flat cylinder, ignoring the presence of villi and microvilli, which significantly increase the effective surface area through which drug absorption can occur. A correction of the absorption area to take into account the presence of the additional surface provided by the villi/microvilli would result in a new estimate for the area, which, by definition, would be greater than that of the smooth cylinder and, hence, reduce the value of *P_eff,human_*. The impact of this additional surface area was recently shown by Olivares-Morales et al., who demonstrated that the estimated *P_eff,human_* changed dramatically upon including a physiologically relevant estimate of the intestinal surface area available for drug absorption [77]. A correction of the surface area decreased *P_eff,human_* and brought the values closer to the permeability coefficients reported ex vivo. The comparison of these corrected *P_eff,human_* and the *P_app,pig_* obtained here using the Ussing chamber system shows excellent agreement (Table 2).

The *P_eff,human_* (uncorrected)/*P_app,pig_* ratios for high permeability drugs were approximately 20-fold greater than those for low permeability drugs. The permeation surface for high permeability drugs, normally absorbed via the transcellular pathway, is larger compared to that of low permeability drugs, which are absorbed by the paracellular pathway. Therefore, a reduction in or a lack of blood flow, motility or lymphatic drainage in ex vivo experiments influences high permeability drugs much more than low permeability drugs—hence, their higher *P_eff,human_* /*P_app,pig_* ratios.

In a next step, the published *P_eff,human_* values were plotted against the ln *P_app, pig_* values for the three experimental setups KBR/KBR, FaSSIF V2/KBR, and FeSSIF V2/KBR (Figure 1). In every case, strong sigmoidal correlations (*R*^2^ = 0.97 for KBR/KBR, *R*^2^ = 0.91 for FaSSIF V2/KBR and *R*^2^ = 0.76 for FeSSIF V2/KBR) were obtained and it was possible to distinguish clearly between high and low permeability drugs. The data were fitted using a four parameter logistic equation (see Section 2.5.2) [28]; in general, *P_app, pig_* decreased going from KBR to FaSSIF V2 and FeSSIF V2, and the correlation curves were translated to the left. The quality of the fit decreased going from KBR/KBR to FeSSIF V2/KBR. A trend towards a decreased permeability with an increasing concentration of sodium taurocholate and lecithin in the biorelevant media was found. This was in good agreement with previous findings, where it was observed that, depending on the drug characteristics, permeability decreased with an increasing concentration of solubility enhancers since poorly water soluble drugs were increasingly solubilized in the micelles, and therefore, less free drug was available for permeation [79,80]. Thus, the composition of FeSSIF V2 makes it more likely to influence the solubility and absorption of certain drugs, and this additional complexity might affect the quality of the correlation.

As the *P_eff,human_* values are obtained for more molecules, further experiments can be carried out with the porcine intestine to determine the corresponding *P_app,pig_* and so to expand the dataset.

### 3.2. Segmental Intestinal Drug Permeation: Regional Variations in Drug Uptake

Physiological factors such as variation of the surface, “tightness” of the tight junctions, variable expression of uptake and efflux transporters as well as enzymes can result in regional differences in drug absorption [81]. The identification of segmental drug permeation differences at an early stage can be advantageous since it can be used to optimize formulation development. Interestingly, the mean transepithelial resistances of the four intestinal segments at the start of the experiments were found to be in the same range: 57.33 ± 20.66 Ω.cm^2^ (*n* = 14) for the duodenum, 41.77 ± 13.79 Ω.cm^2^ (*n* = 155) for the jejunum, 40.85 ± 15.08 Ω.cm^2^ (*n* = 31) for the ileum and 35.71 ± 16.56 Ω.cm^2^ (*n* = 18) for the colon. The ability of the model to identify segmental differences in intestinal drug permeation was tested by determining *P_app,pig_* of antipyrine and metoprolol (two high permeability drugs; BCS I) and cimetidine (low permeability drug; BCS III) across the duodenum, jejunum, ileum and colon (Figure 2 and Table 3). Due to their lipophilicity, antipyrine and metoprolol are absorbed passively via the transcellular pathway [82]. Cimetidine, in contrast, is transported by the paracellular pathway, and it is also a P-gp substrate [83]. The results were compared with the permeability data determined in humans [84].

In the case of antipyrine, *P_app,pig_* across the ileum was significantly greater than that for uptake across the duodenum and jejunum (*p* < 0.05), i.e., an improved permeation on passing from the proximal to the distal small intestine. It was noted that *P_app,pig_* for uptake in the colon was equivalent to that in the ileum. A similar behavior was observed with metoprolol, where a statistically significant increase in *P_app,pig_* from the proximal to the distal gastrointestinal tract was observed (*p* < 0.05). As mentioned above, in contrast to antipyrine and metoprolol, which are BCS I drugs, cimetidine is a BCS III drug, and it behaved differently: In this case, *P_app,pig_* was significantly higher in the jejunum as compared to all the other segments (*p* < 0.05).

To date, few in vivo data regarding a regional variation in drug permeability in humans are available. Of the drugs tested here, *P_eff,human_* has only been determined for cimetidine [85], and the same trend was observed: The *P_eff,human_* across jejunum was significantly higher than that across the other segments (although the absolute value of *P_app,pig_* was approx. 50-fold lower than (uncorrected) *P_eff,human_*). It was found that *P_eff,human_* decreased three-fold going from the jejunum to the ileum (from 75 × 10^−6^ cm/s to 25 × 10^−6^ cm/s), in comparison, *P_app,pig_* decreased approx. eight-fold. 

In terms of ex vivo data from human intestine in the Ussing chamber (*P_app,human_*), (i) for antipyrine, the *P_app,human_* values increased approximately two-fold on going from the duodenum to the colon (26.3 ± 6.99 × 10^−6^ cm/s to 54.6 ± 13.0 × 10^−6^ cm/s) [28] and were similar to *P_app,pig_* measured here (13.7 ± 5.2 × 10^−6^ cm/s and 28.6 ± 4.0 × 10^−6^ cm/s, respectively); (ii) for cimetidine, *P_app,human_* was only determined for the jejunum (3.74 ± 0.47 × 10^−6^ cm/s) and was approximately two-fold higher than *P_app,pig_* (1.50 ± 0.07 × 10^−6^ cm/s); and (iii) for metoprolol, *P_app,pig_* was in the same range as *P_app,human_* in the jejunum (10.64 ± 2.92 × 10^−6^ cm/s vs. 15.9 ± 3.69 × 10^−6^ cm/s, respectively) and in the colon (17.55 ± 5.41 × 10^−6^ cm/s vs.18.8 ± 4.0 × 10^−6^ cm/s, respectively) (Table 3) [28]. In general, there was a good agreement between *P_app,human_* and *P_app,pig_* (although *P_app,human_* was usually a little higher) confirming that healthy porcine intestinal tissue was able to identify regional differences in drug absorption consistent with those reported with human intestine [28].

### 3.3. Demonstrating Activity of the P-gp Efflux Transporter 

The transport of two low permeability drugs, cimetidine and ranitidine (both BCS III), which are known substrates of P-gp [81,83,86], in the jejunum and ileum was investigated in the presence and absence of the P-gp inhibitor verapamil (Figure 3). BCS III substances were chosen since P-gp plays a minimal role in the drug permeation of high permeability drugs (BCS I and II) [87]. No statistically significant difference in *P_app,pig_* was observed for transport across the jejunum in the presence or absence of verapamil (Table 4). In contrast, the presence of verapamil significantly increased *P_app,pig_* in the ileum for both cimetidine and ranitidine. In human intestine, P-pg expression in the proximal intestine is lower than in the distal small intestine [41,84]. The more pronounced effect of verapamil on the absorption in the ileum suggested that the same P-gp distribution might be present in porcine intestine. However, since it is reported that verapamil is a modulator of the organic cation transporter, the effect—if any—of this activity needs to be investigated [88,89].

### 3.4. Demonstrating that CYP3A4 Retains Activity in the Porcine Intestine Ex Vivo

Another crucial process during drug permeation in vivo is the pre-systemic metabolism in the gut wall—principally due to CYP3A4 activity. To test whether CYP3A4 activity was retained in porcine intestine ex vivo, the intestinal permeation of three CYP3A4 substrates, midazolam, tamoxifen and verapamil, was investigated in the presence and absence of clotrimazole, a potent CYP3A4 inhibitor. The *P_app,pig_* of all three substances significantly increased in the presence of clotrimazole (Table 5); furthermore, the amount of each of the principal metabolites—α-hydroxymidazolam, N-desmethyl-tamoxifen, and norverapamil—in the intestinal tissue was significantly decreased (Figure 4).

### 3.5. Relative Contribution of Drug Deposition (Q_DEP_) and Drug Permeation (Q_PERM_) to the Transport Index (TI)

The values of *Q_DEP_* and *Q_PERM_* provide insight at a number of levels (Table 2). First, *Q_DEP_* reports on the fraction of drugs that partitioned from the formulation in the donor compartment into the intestinal tissue but was unable to reach the receiver compartment within the timeframe of the permeation experiment. Thus, it is not taken into account in the calculations that measure the permeability coefficient and, hence, may contribute to the underestimation of intestinal absorption ex vivo. Second, the relative magnitudes of *Q_DEP_* and *Q_PERM_* reflect the effect of the drug physicochemical properties on molecular transport into and across the intestine (Table 1 and Table 2). Highly lipophilic molecules, e.g., midazolam, tamoxifen and verapamil, used in the studies to investigate CYP3A4 activity showed a clear trend towards accumulation within the membrane. In a similar way, the moderately lipophilic character of ketoprofen and piroxicam may explain their retention within the intestinal tissue. Although the P-gp substrates, cimetidine and ranitidine, are significantly more polar, they demonstrated a selectivity for membrane accumulation over permeation; this was tentatively attributed to their capacity to form hydrogen bonds (HB): They both contained five HB acceptor groups in addition to multiple HB donor functions—three in the case of cimetidine and two for ranitidine. The relative values of *Q_DEP_* and *Q_PERM_* for metoprolol and, in particular, atenolol were more difficult to explain. In the case of metoprolol, a combination of modest lipophilicity and HB formation capacity (four HB acceptors and two HB donors) might be sufficient to account for its behavior. Atenolol was more polar but contained four HB acceptors and two HB donors. It was noted that most of the molecules with a propensity to accumulate were P-gp substrates, although whether and/or how this might influence retention is unclear at this point. Regional differences in the relative magnitudes of *Q_DEP_* and *Q_PERM_* were observed: The *Q_PERM_*/*Q_DEP_* ratio was significantly greater in the colon for antipyrine, cimetidine and metoprolol than in the small intestine.

In the case of the CYP3A4 substrates, the presence of clotrimazole only had a statistically significant effect on the *Q_DEP_* and *Q_PERM_* of verapamil (*p* < 0.05 and *p* < 0.05, respectively). As shown in Figure 4, during the permeation experiment, the metabolite norverapamil was formed, which is itself a strong CYP3A4 inhibitor [89]. CYP3A4 inhibition by clotrimazole in conjunction with norverapamil may have helped to increase the amount of verapamil in the tissue, which accumulated due to its lipophilicity.

## 4. Conclusions

The results demonstrated that viable porcine intestine ex vivo could be used in conjunction with an Ussing chamber system to evaluate the effect of physiological conditions on intestinal drug absorption. The comparison of passive drug absorption from KBR and with biorelevant media simulating fasted and fed states was a step forward in trying to predict drug absorption under more physiological conditions. An excellent correlation was observed between *P_app,pig_* and the drug permeability coefficient, *P_eff,human_*, measured in vivo both in terms of the trend and absolute values once a correction was made to take into account the actual surface area for absorption in vivo [77]. The experiments investigating the differences in regional/segmental uptake demonstrated the potential of the model to aid in the rational design of formulations that enabled release in the region with the highest drug permeability. The advantages of using viable tissue were clearly shown in the studies illustrating the impact of P-gp transport and pre-systemic metabolism by CYP3A4 present in the gut wall. From an ethical standpoint, the technique had the advantage in that it did not require the sacrifice of any animals in the laboratory. Furthermore, healthy tissue was sourced from a large omnivorous mammal. In the next phase of the project, the aims are (i) to create a dynamic system that mimics molecular transit through the different compartments of the gastrointestinal tract and which enables the direct evaluation of oral drug dosage forms and (ii) to explore the potential applications in food and nutrition and the elucidation of active compounds present in complex formulations used in traditional medicine.

## Figures and Tables

**Figure 1 pharmaceutics-11-00139-f001:**
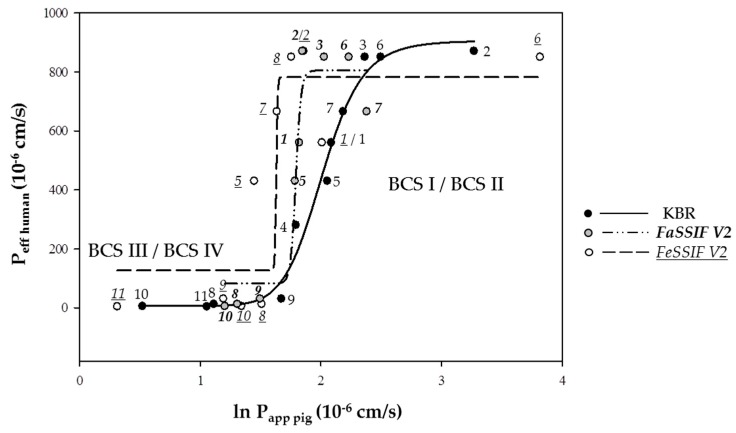
The correlation between the effective permeability coefficient in vivo, *P_eff,human_* (literature values derived assuming smooth cylindrical representation of the surface area for absorption) and the permeability coefficient calculated using porcine intestine ex vivo and the Ussing chamber technique (*P_app,pig_*): The drug absorption was tested using KBR, and the biorelevant media, FaSSIF V2, and FeSSIF V2. BCS I/II (High permeability drugs): 1. Antipyrine, 2. Ketoprofen, 3. Metoprolol, 4. Propranolol, 5. Carbamazepine, 6. Naproxen and 7. Piroxicam; BCS III/IV (Low permeability drugs): 8. Atenolol, 9. Terbutaline, 10. Furosemide and 11. Hydrochlorothiazide (for n, see Table 2). A 100 µM solution of each drug was prepared in (i) KBR, (ii) FaSSIF V2 and (iii) FeSSIF V2. A four parameter logistic model was used to derive the fit between *P_eff,human_* and *P_app,pig_* (see Section 2.5.2) [28].

**Figure 2 pharmaceutics-11-00139-f002:**
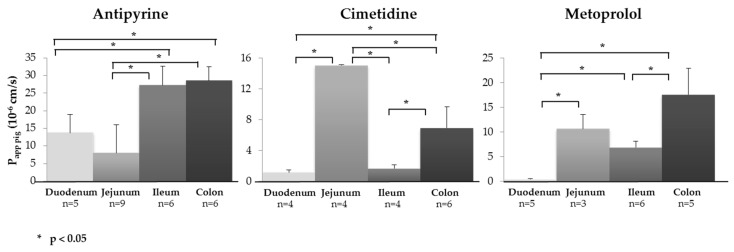
The porcine ex vivo model was able to identify regional variations in intestinal absorption (*P_app,pig_*) for metoprolol, cimetidine and antipyrine in the duodenum, jejunum, ileum and colon. A 100 µM solution of each drug was prepared in KBR (Mean ± SD; *n* = number of replicates). A statistical analysis was performed using one-way ANOVA followed by a Bonferroni’s multiple comparisons ad hoc test.

**Figure 3 pharmaceutics-11-00139-f003:**
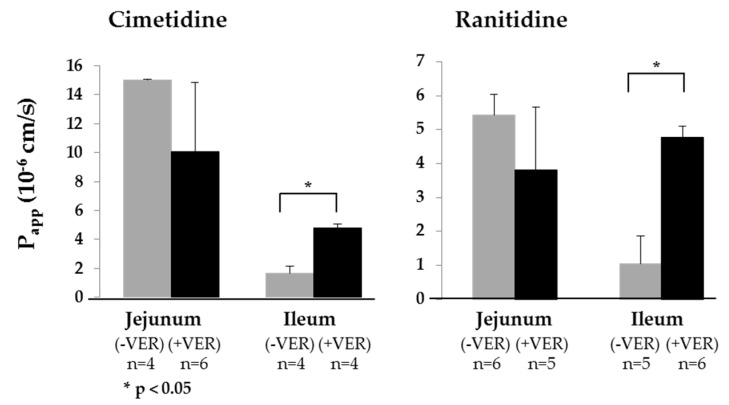
The activity of the P-gp efflux transporter was confirmed by measuring *P_app,pig_* of the P-gp substrates, cimetidine and ranitidine (both prepared at a concentration of 100 µM in KBR), in the jejunum and the ileum in the presence (+VER; 100 µM) or absence (−VER) of verapamil (a P-gp inhibitor). Statistically significant differences (Student’s t-test) were observed for the ileum, which was consistent with the reports for humans in vivo. (Mean ± SD; *n* = number of replicates).

**Figure 4 pharmaceutics-11-00139-f004:**
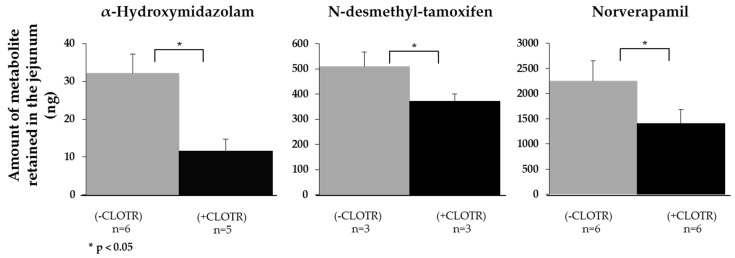
The enzymatic activity of CYP3A4 in the intestinal tissue was confirmed by investigating the transport of the CYP3A4 substrates midazolam, tamoxifen and verapamil (all prepared at a concentration of 100 µM in KBR) across the jejunum and quantification of the amounts (of their metabolites (α-hydroxymidazolam, N-desmethyl-tamoxifen and norverapamil) retained in the tissue in the presence (+CLOTR; 100 µM) or absence (−CLOTR) of the CYP3A4 inhibitor clotrimazole. Statistically significant differences were identified using the Student’s t-test. (Mean ± SD; *n* = number of replicates).

**Table 1 pharmaceutics-11-00139-t001:** The chemical structures and physicochemical properties of the drug molecules tested using the Ussing chamber system.

API	MW (g/mol)	log *P*	log *D* [59] Octanol/H_2_O pH 7.4 pH 6.5 pH 5.5			Solubility (*n* = 3) (mg/mL)			Study
						KBR	FaSSIF V2	FeSSIF V2	
**BCS I**									
Antipyrine (1) C_11_H_12_N_2_O	188.23	0.38	0.6	0.6	0.6	103.00 ± 54.61	261.10 ± 132.29	564.44 ± 83.12	Passive *^b^* Regional *^c^*
Ketoprofen (2) C_16_H_14_O_3_	254.28	3.12	0.1	0.8	1.8	2.82 ± 0.62	6.73 ± 2.70	6.37 ± 1.52	Passive
(+/−)-Metoprolol (3) C_15_H_25_NO_3_	267.36	1.88	0.0	−0.5	−0.6	298.03 ± 13.86	43.85 ± 15.81	380.69 ± 33.79	Passive Regional
Midazolam C_18_H_13_ClFN_3_	325.77	3.89	3.0 *^a^*	3.6 *^a^*	3.9 *^a^*	n.a.	n.a.	n.a.	CYP3A4 *^d^*
Propranolol (4) C_16_H_21_NO_2_	259.34	3.48	1.4	0.9	0.7	158.69 ± 4.34	247.53 ± 7.47	222.57 ± 2.00	Passive
Verapamil C_27_H_38_N_2_O_4_	454.60	3.79	3.8*^a^*	3.8 *^a^*	3.8 *^a^*	1.04 ± 0.09	5.60 ± 1.04	17.60 ± 0.98	CYP3A4
**BCS II**									
Carbamazepine (5) C_15_H_12_N_2_O	236.27	2.1	2.45	2.45	2.45	0.20 ± 0.02	0.32 ± 0.01	0.73 ± 0.01	Passive
Naproxen (6) C_14_H_14_O_3_	230	3.18	0.3	1.1	2.1	4.51 ± 0.04	14.59 ± 1.19	13.00 ± 2.37	Passive
Piroxicam (7) C_15_H_13_N_3_O_4_S	331.35	3.06	0.2 *^a^*	1.1 *^a^*	2.0 *^a^*	0.43 ± 0.02	0.29 ± 0.02	0.06 ± 0.00	Passive
Tamoxifen C_26_H_29_NO	371.51	5.93	5.6 *^a^*	5.9 *^a^*	5.9 *^a^*	23.32 ± 1.54	32.19 ± 0.42	27.87 ± 0.66	CYP3A4
**BSC III**									
Atenolol (8) C_14_H_22_N_2_O_3_	365.40	0.75	−2.0	−2.0	−2.0	35.57 ± 7.29	33.73 ± 2.06	41.28 ± 1.16	Passive
Cimetidine C_14_H_22_N_2_O_3_	252.34	0.40	0.4 *^a^*	0.4 *^a^*	0.4 *^a^*	5.65 ± 0.13	5.88 ± 0.26	0.36 ± 0.26	Regional P-gp *^e^*
Ranitidine C_14_H_22_N_2_O_3_	314.40	0.27	0.2 *^a^*	0.3 *^a^*	0.3 *^a^*	19.64 ± 2.78	15.89 ± 1.47	807.51 ± 70.47	P-gp
Terbutaline (9) C_14_H_22_N_2_O_3_	225.28	0.9	−1.4	−1.3	−1.3	213.73 ± 15.18	372.59 ± 18.86	305.07 ± 124.48	Passive
**BCS IV**									
Furosemide (10) C_12_H_11_ClN_2_O_5_S	330.74	2.03	−0.9	−0.5	0.4	5.62 ± 0.10	24.90 ± 2.84	19.68 ± 2.59	Passive
Hydrochlorothiazide (11) C_7_H_8_ClN_3_O_4_S_2_	297.74	−0.16	−0.2	−0.2	−0.2	0.81 ± 0.17	1.18 ± 0.14	1.04 ± 0.19	Passive

*^a^* The log *D* value was not taken from Reference [54] but was calculated using the following equation: $\log D=\log P+\log(\frac{1}{1+10^{pH-pKa}});$
*^b^* Passive: the drug used to study passive drug permeation from KBR and biorelevant media; *^c^* Regional: the drug used to study drug permeation in different intestinal segments; *^d^* CYP3A4: the drug used to study CYP3A4 activity; *^e^* P-gp: the drug used to study P-gp activity.

**Table 2 pharmaceutics-11-00139-t002:** The apparent permeability coefficients for passive absorption across porcine intestine, *P_app,pig_*, from KBR and the biorelevant media FaSSIF V2 and FeSSIF V2 and the *Q_DEP_* and *Q_PERM_* values determined with viable porcine intestine ex vivo (n ≥ 3): The effective permeability coefficients^a,b^ for absorption in vivo in humans, *P_eff, human_*, are given for comparison.

Drug	*P_app,pig_* Ex Vivo (10^−6^ cm/s)			*P_eff,human_^a^*	*P_eff,human_^b^*				*Q_DEP_* (%)	*Q_PERM_* (%)	(*n*) *^c^*			
	KBR (*n*)	FaSSIF V2 (*n*)	FeSSIF V2 (*n*)	In Vivo (10^−6^ cm/s)	In Vivo (10^−6^ cm/s)	KBR			FaSSIF V2			FeSSIF V2		
**BCS I**														
(1) Antipyrine	8.06 ± 7.91 (10)	6.18 ± 2.18 (4)	7.47 ± 0.95 (3)	560 ± n.a. ^[31]^	19–29 ^[77]^	0.72 ± 0.20	2.18 ± 0.53	(6)	1.07 ± 0.22	0.33 ± 0.16	(4)	2.90 ± 0.26	1.03 ± 0.28	(3)
(2) Ketoprofen	26.31 ± 1.49 (3)	6.34 ± 2.63 (5)	6.42 ± 2.44 (4)	870 ± n.a. ^[31]^	29–45 ^[77]^	11.57 ± 2.65	6.25 ± 0.92	(6)	4.49 ± 1.61	0.90 ± 0.46	(5)	0.37 ± 0.08	0.97 ± 0.54	(4)
(3) Metoprolol	10.64 ± 2.92 (3)	7.62 ± 1.41 (3)	5.79 ± 1.38 (3)	850 ± n.a. ^[31]^	5.2–7.9 ^[77]^	0.28 ± 0.04	0.18 ± 0.05	(6)	0.12 ± 0.03	0.89 ± 0.19	(3)	0.22 ± 0.01	0.74 ± 0.08	(3)
(4) Propranolol	6.01 ± 3.41(5)	0.71 ± 0.24 (3)	0.93 ± 0.55 (3)	280 ± 130 ^[31]^	9.3–14 ^[77]^	0.35 ± 0.06	0.04 ± 0.05	(6)	4.17 ± 0.15	0.27 ± 0.03	(3)	1.61 ± 0.57	0.27 ± 0.30	(3)
**BCS II**														
(5) Carbamazepine	7.81 ± 5.69 (16)	5.97 ± 0.52 (4)	4.26 ± 0.96 (6)	430 ± n.a. ^[31]^	-	1.44 ± 0.16	2.39 ± 0.42	(6)	n.a.	n.a.	-	3.45 ± 0.52	0.58 ± 0.22	(6)
(6) Naproxen	12.14 ± 4.83 (3)	9.33 ± 3.36 (3)	45.47 ± 7.97 (5)	850 ± n.a. ^[31]^	27–42 ^[77]^	n.a.	n.a.	-	n.a.	n.a.	-	4.74 ± 1.04	8.11 ± 1.22	(5)
(7) Piroxicam	8.93 ± 2.03 (3)	10.83 ± 3.70 (3)	5.14 ± 0.78 (4)	665 ± n.a. ^[31]^	-	4.12 ± 0.15	2.52 ± 0.70	(3)	5.63 ± 0.26	2.32 ± 0.79	(3)	0.21 ± 0.04	1.13 ± 0.46	(5)
**BCS III**														
(8) Atenolol	3.05 ± 0.73 (6)	3.70 ± 1.28 (6)	4.53 ± 1.71 (5)	20 ± n.a. ^[31]^	1.8–2.8 ^[77]^	7.38 ± 3.11	0.67 ± 0.13	(6)	0.25 ± 0.06	0.48 ± 0.09	(6)	0.55 ± 0.11	0.61 ± 0.12	(5)
(9) Terbutaline	5.33 ± 1.95 (17)	4.47 ± 0.62 (5)	3.30 ± 0.68 (6)	30 ± 30 ^[78]^	1.7–2.6 ^[77]^	0.52 ± 0.11	0.66 ± 0.22	(6)	39.47 ± 2.51	0.54 ± 0.29	(5)	0.04 ± 0.00	0.39 ± 0.10	(6)
**BCS IV**														
(10) Furosemide	1.69 ± 0.89 (16)	3.34 ± 1.46 (4)	3.83 ± 0.77 (6)	5 ± n.a. ^[31]^	1–1.6 ^[77]^	n.a.	n.a.	-	2.22 ± 0.17	0.43 ± 0.09	(4)	1.52 ± 0.24	1.38 ± 0.24	(6)
(11) Hydrochlorothiazide	2.88 ± 2.84 (17)	0.2 ± 0.25 (3)	1.37 ± 0.53 (6)	4 ± n.a. ^[31]^	-	n.a.	n.a.	-	0.22 ± 0.11	0.11 ± 0.05	(3)	0.74 ± 0.09	0.23 ± 0.08	(6)

*^a^ P_eff,human_* determined using the Loc-I-Gut method was performed in living humans. A jejunal segment of 10 cm was isolated and perfused with a solution with a known drug concentration. The difference in drug concentrations across the isolated segment was used to estimate the drug absorption, and this amount together with the assumption of a cylindrical surface area for uptake enabled the estimation of *P_eff,human_*; *^b^ P_eff,human_* calculated using the Loc-I-Gut method was performed in living humans but with the inclusion of physiological estimates of the surface area for absorption [77]; *^c^* Number of replicates.

**Table 3 pharmaceutics-11-00139-t003:** The apparent drug permeability coefficients, *P_app,pig_* and *Q_DEP_* and *Q_PERM_*, for drugs in different segments of the small intestine (duodenum, jejunum and ileum) and the colon.

Drug	*P_app,pig_* Ex Vivo (10^−6^ cm/s)				*Q_DEP_* (%) *Q_PERM_* (%) (*n*) *^a^*							
	Duodenum (*n*)	Jejunum (*n*)	Ileum (*n*)	Colon (*n*)	Duodenum		Jejunum		Ileum		Colon	
Antipyrine	13.69 ± 5.24 (5)	8.06 ± 7.91 (9)	27.26 ± 5.47 (6)	28.56 ± 3.97 (6)	1.10 ± 0.47	1.57 ± 0.58 (5)	0.72 ± 0.20	2.18 ± 0.53 (5)	1.97 ± 0.34	1.39 ± 0.56 (6)	0.87 ± 0.16	3.79 ± 0.80 (5)
Cimetidine	1.12 ± 0.30 (4)	15.01 ± 0.07 (4)	1.66 ± 0.48 (4)	6.91 ± 2.74 (6)	0.67 ± 0.06	0.91 ± 0.27 (4)	3.28 ± 0.56	1.66 ± 0.64 (6)	0.07 ± 0.01	0.26± 0.08 (3)	0.05 ± 0.01	0.81 ± 0.25 (6)
Metoprolol	0.39 ± 0.16 (5)	10.64 ± 2.92 (3)	6.87 ± 1.23 (6)	17.55 ± 5.41 (6)	2.35 ± 0.25	0.14 ± 0.08 (5)	0.27 ± 0.04	0.18 ± 0.05 (4)	1.89 ± 0.67	0.83 ± 0.09 (5)	0.02 ± 0.00	1.65 ± 0.87 (6)

^a^ Number of replicates.

**Table 4 pharmaceutics-11-00139-t004:** The apparent drug permeability coefficients, *P_app,pig_* and *Q_DEP_* and *Q_PERM_*, in the jejunum and ileum for the drug substrates of the P-gp efflux transporter in the presence (+VER) and absence (−VER) of the P-gp inhibitor, verapamil.

Drug	*P_app,pig_* Ex Vivo (10^−6^ cm/s)				*Q_DEP_* (%) *Q_PERM_* (%) (*n*) ^*a*^							
	Jejunum		Ileum		Jejunum		Jejunum		Ileum		Ileum	
	(−VER) (*n*)	(+VER) (*n*)	(−VER) (*n*)	(+VER) (*n*)	(−VER)		(+VER)		(−VER)		(+VER)	
Cimetidine	15.01 ± 0.07 (4)	10.07 ± 4.80 (6)	1.66 ± 0.48 (4)	4.28 ± 0.70 (4)	2.98 ± 0.29	1.38 ± 0.03 (4)	4.32 ± 0.97	1.49 ± 0.67(6)	0.07 ± 0.01	0.26 ± 0.08 (3)	0.08 ± 0.02	0.71 ± 0.14 (4)
Ranitidine	5.42 ± 0.61 (6)	4.92 ± 0.30 (5)	1.04 ± 0.83 (5)	4.78 ± 0.31 (6)	1.32 ± 0.31	0.67 ± 0.12 (6)	5.47 ± 0.75	0.23 ± 0.15 (5)	1.10 ± 0.59	0.55 ± 0.35 (4)	7.91 ± 0.74	0.25 ± 0.03 (6)

*^a^* Number of replicates.

**Table 5 pharmaceutics-11-00139-t005:** The apparent drug permeability coefficients, *P_app, pig_* and *Q_DEP_* and *Q_PERM_*, in the jejunum for drug substrates of CYP3A4 in the presence (+CLOTR) and absence (−CLOTR) of the CYP3A4 inhibitor, clotrimazole.

Drug	*P_app,pig_* Ex Vivo (10^−6^ cm/s)				*Q_DEP_* (%) *Q_PERM_* (%) (*n*) ^*a*^					
	(−CLOTR)	(*n*)	(+CLOTR)	(*n*)	(−CLOTR)			(+CLOTR)		
Midazolam	0.183 ± 0.138	(4)	0.460 ± 0.070	(4)	8.06 ± 1.33	0.13 ± 0.09	(6)	7.49 ± 2.50	0.18 ± 0.08	(5)
Tamoxifen	0.124 ± 0.046	(3)	1.381 ± 1.080	(3)	1.82 ± 1.39	0.07 ± 0.07	(3)	1.82 ± 0.58	0.06 ± 0.06	(4)
Verapamil	0.008 ± 0.003	(4)	0.211 ± 0.071	(3)	1.88 ± 0.12	0.00 ± 0.00	(3)	3.00 ± 0.63	0.18 ± 0.07	(6)

*^a^* Number of replicates.

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
