# Peer review of "Drug Transport across Porcine Intestine Using an Ussing Chamber System: Regional Differences and the Effect of P-Glycoprotein and CYP3A4 Activity on Drug Absorption"

_pharmaceutics, 2019, doi:10.3390/pharmaceutics11030139_

Round 1

Reviewer 1 Report

This manuscript describes the permeation of drugs from different BCS classes across excised pig intestinal tissues mounted in an Ussing type chamber system and compared it to each drug’s permeation in humans.  The study also investigated the effect of the region of the gastro-intestinal tract as well as P-gp efflux and metabolism on drug permeation.  The drug permeated across the excised pig intestinal tissues was measured in addition to the amount of drug retained in the tissues that relates to “delivery efficiency”.  This is an excellent study and the manuscript is well written with high value to readers in this field.  It fills a gap in the knowledge of ex vivo permeation across pig intestinal tissue and is very valuable.  The following minor questions/recommendations should be addressed:

1)  Page 3, line 95:  Other studies that used excised pig intestinal tissues in Ussing type chamber experiments should also be cited here such as Aucamp et al., 2015 (Drug Dev Ind Pharm, 41:1100-118), Atlabachew et al., 2016 (Journal of Ethnopharmacology, 194:307-315), De Bruyn et al., 2018 (Drug Delivery Letters, 8:52-60) and Gerber et al., 2018 (Journal of Food and Drug Analysis, 26:S115-S124).

2)  Page 3, line 101:  Two other studies that investigated excised pig intestinal tissues for drug permeation studies should be acknowledge here namely the work by Neirinckx et al., 2010 in Journal of Veterenary Pharmacology and Therapeutics, 34:290-297 - they have conducted a study to compare excised intestinal tissues from different animals (including pig) in terms of drug permeation and histology.  Pietzonka et al 2002 in Eur J Pharm Sci, 15:39-47 also investigated different aspects of excised pig intestinal tissues as permeation model in Ussing type chambers.

3)  Page 4, line 146:  Was the thickness of the tunica mucosa and submucosa measured after removal of the muscle layer?  How did the thickness varied between the different experiments?

4)  Page 4, line 163:  It is stated that the tissue was mounted 45 min after isolation from the animal, but how long was this after death (or slaughter) of the animal? (in other words what was the time from death to tissue isolation?).

5)  Page 11, line 195:  How was tissue integrity determined during the preliminary studies? E.g. was transport of an exclusion marker also measured in addition to TEER?

6)  Page 12, line 224:  Why was dc/dt only calculated between 20 and 80 min, while the permeation studies were conducted over 120 min?

7)  Page 12, lines 249-250:  Was data tested for normal distribution to ensure ANOVA is appropriate?  Why was a post-hoc test not applied?

Author Response

Reviewer 1

1)  Page 3, line 95:  Other studies that used excised pig intestinal tissues in Ussing type chamber experiments should also be cited here such as Aucamp et al., 2015 (Drug Dev Ind Pharm, 41:1100-118), Atlabachew et al., 2016 (Journal of Ethnopharmacology, 194:307-315), De Bruyn et al., 2018 (Drug Delivery Letters, 8:52-60) and Gerber et al., 2018 (Journal of Food and Drug Analysis, 26:S115-S124).

The original references only cited studies where Ussing chambers were used. The articles mentioned by the reviewer have been added to the manuscript (Refs 50-53) and the text modified to include “…Ussing chamber or related systems and with diffusion cells…”. (Edited text highlighted (lines 93-97)).

2)  Page 3, line 101:  Two other studies that investigated excised pig intestinal tissues for drug permeation studies should be acknowledge here namely the work by Neirinckx et al., 2010 in Journal of Veterenary Pharmacology and Therapeutics, 34:290-297 - they have conducted a study to compare excised intestinal tissues from different animals (including pig) in terms of drug permeation and histology.  Pietzonka et al 2002 in Eur J Pharm Sci, 15:39-47 also investigated different aspects of excised pig intestinal tissues as permeation model in Ussing type chambers.

The study of Neirinckx et al. was already cited (Ref. 46). The article of Pietzonka et al. has been added (Ref 54).

3)  Page 4, line 146:  Was the thickness of the tunica mucosa and submucosa measured after removal of the muscle layer?  How did the thickness varied between the different experiments?

The thickness of the tunica mucosa and submucosa was not measured and this has not been done in the vast majority of studies published to date. Inter-animal variability was investigated in a set of preliminary experiments where Papp of propranolol, a drug absorbed by passive diffusion, was determined using intestine from different animals and the Papp values compared. The results showed that there was no significant difference between the Papp derived using the intestine from the different pigs.

4)  Page 4, line 163:  It is stated that the tissue was mounted 45 min after isolation from the animal, but how long was this after death (or slaughter) of the animal? (in other words what was the time from death to tissue isolation?).

Time from slaughter of the animal to tissue isolation was 5 to 10 min - at which point it was stored in the cooled, oxygenated KBR. The 45 min refers to the total time between the harvesting of the intestine and the insertion of the tunica mucosa into the Ussing chamber inserts. Edited text highlighted (line 163).

5)  Page 11, line 195:  How was tissue integrity determined during the preliminary studies? E.g. was transport of an exclusion marker also measured in addition to TEER?

Tissue integrity was determined by continuous monitoring of TEER throughout the experiment (as described from line 192-199 in the manuscript). This technique is well known and very reliable. Therefore, an exclusion marker was not used, thereby also avoiding possible interactions between the marker and the test drug.

6)  Page 12, line 224:  Why was dc/dt only calculated between 20 and 80 min, while the permeation studies were conducted over 120 min?

Preliminary, histological studies showed that the intestine started to display signs of degradation between 100 and 120 min. Therefore, it was decided to calculate Papp from the slope of the concentration – time curve between 20 and 80 min when integrity of the membrane was ensured.

7)  Page 12, lines 249-250:  Was data tested for normal distribution to ensure ANOVA is appropriate?  Why was a post-hoc test not applied?

Normal distribution was assumed. Bonferroni’s ad hoc test was carried out following the ANOVA. The information has been added to the Materials and Methods (section 2.6 - (Edited text highlighted (line 249))) and to the legend for Figure 2 (Edited text highlighted (line 351-353)). Furthermore, details of the statistical tests used have also been added to the legends for Figure 3 and 4 (Edited text highlighted (lines 401 and 423)).

Reviewer 2 Report

The submitted manuscript “Drug transport across porcine intestine using an Ussing chamber system: regional differences and effect of P-glycoprotein and CYP3A4 activity on drug absorption” by Kalia et al demonstrated an ex vivo system for drug absorption by using Ussing chamber system, which enabled to perform transport experiments on viable intestinal tissue. They have demonstrated the porcine intestine can be used as a surrogate to study drug intestinal absorption ex vivo with 11 BCS drugs. With the inhibitors of P-gp (verapamil) and CYP3A4 (clotrimazole), they showed the advantages with viable tissues instead of the sacrifice of any animals for the P-gp efflux transporter and CYP3A4 activity. According to their study, the Ussing chamber system will be a good way for high-throughput methods for permeability screening. Therefore, I recommend this manuscript to be published on Pharmaceutics without revision.

Author Response

Reviewer 2

We thank the reviewer for the comments